# The Elaboration of miRNA Regulation and Gene Regulatory Networks in Plant–Microbe Interactions

**DOI:** 10.3390/genes10040310

**Published:** 2019-04-21

**Authors:** Sophie de Vries, Jan de Vries, Laura E. Rose

**Affiliations:** 1Department of Biochemistry and Molecular Biology, Dalhousie University, Halifax, NS B3H 4R2, Canada; jan.devries@dal.ca; 2Institute of Microbiology, Technische Universität Braunschweig, Braunschweig, 38106 Braunschweig, Germany; 3Institute of Population Genetics, Heinrich-Heine-University Düsseldorf, 40225 Düsseldorf, Germany; 4CEPLAS—Cluster of Excellence in Plant Sciences, Heinrich-Heine University Duesseldorf, 40225 Duesseldorf, Germany

**Keywords:** plant evolution, gene expression, co-evolution, plant pathogens, plant–microbe interaction, molecular plant pathology, plant immunity

## Abstract

Plants are exposed to diverse abiotic and biotic stimuli. These require fast and specific integrated responses. Such responses are coordinated at the protein and transcript levels and are incorporated into larger regulatory networks. Here, we focus on the evolution of transcriptional regulatory networks involved in plant–pathogen interactions. We discuss the evolution of regulatory networks and their role in fine-tuning plant defense responses. Based on the observation that many of the cornerstones of immune signaling in angiosperms are also present in streptophyte algae, it is likely that some regulatory components also predate the origin of land plants. The degree of functional conservation of many of these ancient components has not been elucidated. However, ongoing functional analyses in bryophytes show that some components are conserved. Hence, some of these regulatory components and how they are wired may also trace back to the last common ancestor of land plants or earlier. Of course, an understanding of the similarities and differences during the evolution of plant defense networks cannot ignore the lineage-specific coevolution between plants and their pathogens. In this review, we specifically focus on the small RNA regulatory networks involved in fine-tuning of the strength and timing of defense responses and highlight examples of pathogen exploitation of the host RNA silencing system. These examples illustrate well how pathogens frequently target gene regulation and thereby alter immune responses on a larger scale. That this is effective is demonstrated by the diversity of pathogens from distinct kingdoms capable of manipulating the same gene regulatory networks, such as the RNA silencing machinery.

## 1. Introduction

Plants face a plethora of different environmental stimuli throughout their lifetimes. These can originate from abiotic and biotic sources and both can induce changes in plant gene and protein regulation. At the molecular level, these different stress responses often have one thing in common: Many are inducible. Following the perception of the stimulus (biotic or abiotic), time is of the essence. Velocity and efficiency in the stress response must be balanced through modifications at the protein and transcript levels. The latter results in characteristic shifts in global transcriptional profiles associated with certain stimuli (e.g., Reference [1,2,3]). Hence, a plant stress response involves many highly coordinated transcriptional regulatory networks that are organized within a hierarchical structure [4,5,6].

The fundamental role of transcriptional rewiring for a coordinated stress response is reflected in the manifold ways that successful plant pathogens manipulate and alter plant transcription. Effector proteins that are secreted by pathogens to alter plant immune responses [7] can act by modifying chromatin [8,9], suppressing RNA silencing [10,11,12], manipulating plant transcription factors [13], or encoding transcription factors that directly act on the plant chromosomes [14,15,16]. Furthermore, some pathogens secrete small RNA (sRNA) molecules that mimic endogenous plant sRNAs and target plant genes [17,18]. Altering transcription can allow pathogens not only to downregulate specific immunity-associated genes but also to dysregulate multiple pathways simultaneously or to reprogram transcript investments away from immunity-associated pathways to growth-associated metabolism. That these mechanisms are effective is evident when looking across the diversity of pathogens—similar plant regulatory networks are often targeted convergently by pathogens of different kingdoms [19,20,21]. This prevalence of convergent evolution by diverse pathogens to target common plant regulatory networks may emerge because the core of some stress response pathways is conserved across the entire plant tree of life [21].

Here, we will first provide a broader overview of the origin and evolution of stress-associated networks in streptophyte algae and land plants with a focus on key biotic (pathogenic and symbiotic) signaling networks. Second, using the RNA silencing machinery as a prime example, we will highlight the role of gene regulation for an effective immune response. Furthermore, we will discuss how pathogens can exploit and shape such gene regulatory mechanisms to their own advantage.

## 2. Evolution of Plant–Microbe Interaction Networks

Land plants (embryophytes) are a monophyletic group that is estimated to be 474 to 515 million years old [22]. The entire clade is deeply split into the bryophytes and tracheophytes (Figure 1); whether the bryophytes are monophyletic is still debated, but the assertion of monophyly receives more and more support [23,24,25]. The bryophytes encompass the mosses, liverworts, and hornworts (Figure 1). The tracheophytes encompass the lycophytes and euphyllophytes; the latter can be broken up into ferns and seed plants. Within the seed plants are the gymnosperms (e.g., conifers) and angiosperms (Figure 1).

Our understanding of the molecular biology of plants is extremely biased, with a main emphasis on a limited set of angiosperm lineages. In fact, the first complete genomes of ferns have been published only a few months ago (*Azolla filiculoides* and *Salvinia cucullata*; [26]). Most experimental work has been carried out on flowering plants, foremost in the plant model system *Arabidopsis thaliana* [27]. Fortunately, work on additional lineages is rapidly advancing [28,29]. One bryophyte species that is already among the classics is the moss *Physcomitrella patens* [30,31,32]. Additionally, the liverwort *Marchantia polymorpha* is gaining more and more attention and usage [33]. Comparative analyses of these two bryophytes with flowering plants has allowed scientists to retrace the evolutionary history back more than 500 million years. One major insight from these studies is that a large part of the complex transcriptional regulatory cascades of land plants was already present in their most recent common ancestors and even earlier [34,35].

Many of the molecular circuits that connect the perception of the environment to the appropriate transcriptional response are as ancient as land plants—and often likely older since they are found in the closest algal relatives of land plants, the streptophyte algae [36,37,38]. These ancient environmental response circuits include, for example, the signaling based upon two plant hormones: (i) abscisic acid (ABA), which was present in the last common ancestor of land plants [39,40,41]—the respective genes for it might have even deeper roots than initially thought, see [42]—as well as (ii) salicylic acid (SA; see our recent overview and synthesis in Reference [43]). Indeed, the SA receptor homolog Non-expressor of Pathogenesis-Related gene 1 (NPR1) of *P. patens* can partially complement the immunity phenotype of the *npr1* mutant of *A. thaliana* [44]. However, additional experiments are required to further determine the functionality of NPR1 homologs in immunity in bryophytes. Additionally, various genes for the phytohormone signaling, including the biotic stress hormones jasmonic acid (JA) and ethylene (ET), have been found in the genomes of streptophyte algae [45,46,47,48], as well as other stress-metabolite biosynthesis pathways, such as the phenylpropanoid pathway [49]. The machinery and wiring for the JA-signaling pathway are conserved in angiosperms and the liverwort *M. polymorpha*, although these plants rely on different oxylipin ligands to set the signaling cascade into motion [50,51].

Furthermore, nearly the entire signaling module for establishing symbiotic interactions with arbuscular mycorrhizae is inferred to have been present in a common ancestor of land plants and streptophyte algae [52]. This study [52] also found one receptor-like kinase (RLK) in the transcriptomic data of several streptophyte algae. RLKs are key players in land plants for distinguishing between beneficial and antagonistic microbes [53]. New insights about the existence of these genes and their functions can be drawn from comparative genomics. In *Chara braunii* (a charophyceaen fresh water streptophyte alga), the LysM-RLK family appears to have undergone a lineage-specific expansion [48]. Since this class of genes is encoded by a large gene family in most land plants, these independent expansions may be driven by convergent evolution. The extent to which downstream components of these ancient modules are functionally interchangeable can be tested by allelic complementation. Using this technique, Delaux and colleagues [52] could demonstrate that the calcium- and calmodulin-dependent protein kinase (CCaMK) from *Closterium peracerosum*, a zygnematophycean streptophyte alga, could complement a knockout of this gene in *Medicago truncatula.* However, some of the additional downstream components of the pathway in *Medicago* were not present in the algal transcriptomic and genomic data (e.g., those specifying the root cell differentiation during symbiosis). Therefore, some, but not all, components of the symbiosis pathway can be inferred as having been present in the earliest land plants. However, whether extant streptophyte algae use divergent downstream signaling proteins and what this implies for the evolution of symbiosis signaling and the functional modules required for such symbiosis remains an open question.

Not only components of the pathways that are involved in beneficial interactions with symbiotic microbes but also nucleotide-binding site-leucine-rich repeats (NBS-LRRs) are present in the closest algal relatives of land plants. NBS-LRRs are proteins that can recognize pathogen-secreted effector molecules or their function [54]. They are likely present in all major land plant lineages because they have been found in bryophytes, lycophytes, gymnosperms, and *Amborella* (the sister lineage to all angiosperms) and angiosperms [55,56]. A recent study reported the presence of NBS-LRRs in several streptophyte algae [56]. Furthermore, Shao and colleagues [57] found *NBS-LRR*-like sequences in the genomes of chlorophytes, yet because of their scattered distribution in these lineages, Han [58] suggested that, for now, an origin in the ancestor of streptophytes should be assumed.

## 3. The Evolution of microRNA-Mediated Regulation of Resistance Genes

Biotic stress responses need to be tightly regulated and should not be activated in the absence of pathogens. As a consequence, many defense responses are under some form of negative regulation. However, although some *NBS-LRRs* are constitutively expressed, their protein-products are not constitutively active. This can be achieved in two ways: (i) NBS-LRR proteins can undergo conformational changes upon recognition of a pathogen, thereby adopting their active forms [59], or (ii) *NBS-LRRs* can be posttranscriptionally regulated at the mRNA level.

The abundance of *NBS-LRR* transcripts can be regulated by microRNAs (miRNAs) [60,61]. The biogenesis of sRNAs starts with either primary miRNA (pri-miRNA)-encoding genes in the genome or long double-stranded RNA (dsRNA) molecules. A Dicer (DCL) binds the pri-miRNA hairpin structure [62] or long dsRNA molecules [63] and cleaves them into smaller dsRNA pieces. The small dsRNA molecules dissociate into small single-stranded RNA molecules, sRNAs [64]. sRNAs are then incorporated into Argonaute proteins (AGO) which direct them to their complementary mRNA sequences [64,65]. This mechanism leads to one of three possible outcomes, all of which can occur when *NBS-LRRs* are targeted by miRNAs: (i) The mRNA is degraded (posttranscriptional regulation; [61,66]), (ii) the translation of the mRNA is inhibited (translational regulation; [67]), or (iii) a RNA-dependent RNA polymerase 6 (RDR6) is recruited to the mRNA. RDR6 produces long dsRNA molecules from the target mRNA, which are cleaved into secondary small interfering RNAs (siRNAs) [68,69,70,71,72]. These secondary siRNAs are produced in a phased pattern of 21 nucleotides and are named phased, secondary, small interfering RNAs (phasiRNAs) [60,61,68,69]. 

*NBS-LRRs* are regulated via both miRNAs and phasiRNAs [61,66,67]. One well-characterized miRNA family is miR482/2118. Small RNAs encoded by the miR482/2118 family target the P-loop-encoding sequence in NBS-LRR-encoding transcripts [61]. As the P-loop-encoding sequence is omnipresent in *NBS-LRRs*, this allows for a wide-spread regulation of *NBS-LRRs* by only a few miR482/2118 variants. In tomato and potato, roughly 20% of the *NBS-LRR* repertoire is predicted to be targeted by this miRNA family alone [73]. 

Across the land plant lineage, miR482/2118-like sequences were found in both angiosperms and gymnosperms [60,61,73] but not in other lineages. This suggests that the ancestral miR482/2118 gene arose at the base of seed plants (i.e., in the ancestor of gymnosperms and angiosperms; Figure 1). One mechanism by which miRNAs are hypothesized to have originated is from an inverted duplication of the coding sequences of their target genes [74]. The transcription of the inverted duplication could result in a novel RNA molecule which, due to its complementarity, forms a hairpin construct. This hairpin structure could then be further processed into a novel miRNA. The successive sequence evolution of this gene may eventually lead to the origin of a functional miRNA. Consistent with the theory of miRNA origin, comparative genomics has identified pri-miRNAs (the miRNA transcripts before processing) in the genome of *Picea abies* with a sequence similarity to NBS-LRR-encoding genes, specifically in regions outside of the miRNA/miRNA* region [75]. The miRNA/miRNA* is the dsRNA region excised from the precursor miRNA (pre-miRNA) that includes the functional miRNA itself and its complementary sequence denoted as miRNA*. In contrast, no evidence of sequence similarity between pri-miRNAs of the miR482/2118 family and *NBS-LRR* genes outside of the miRNA/miRNA* could be found in tomato [73]. This implies that a continued evolutionary change in these pri-miRNAs present in tomato has erased the telltale signs of their putative origins.

The tissue and target specificity of the miR482/2118 genes may also give insights into the origin of the miR482/2118 family. In gymnosperms, the gene family members are expressed in both vegetative and reproductive tissue, with a subset enriched in male cones [75]. In monocots, the gene family targets noncoding RNAs in reproductive tissue and regulates flower organ development [76], while in dicots, the gene family regulates *NBS-LRRs* [60,61,77]. In summary, the miR482/2118 genes from *P. abies* display a broader expression across tissues, while a greater specialization for miR482/2118 genes in terms of tissue and target specificity is found in the angiosperm lineages. However, in this context, it is important to note that a recent paper reports that one member of miR482/2118 in tomato is exclusively expressed in reproductive tissue [78]. This could mean that some miR482/2118 genes in angiosperms have retained the presumed ancestral dual tissue-specific expression. However, further research is necessary to understand the function, targets, and tissue-specificity of the different family members and how broadly these patterns are conserved across other dicots.

Although *NBS-LRR* genes are found outside of seed plants, their regulation by this miRNA family seems to be restricted to seed plants. In the seed plants, the *NBS-LRR* gene family has expanded independently in different plant lineages [79]. Its first sudden and large expansion may have occurred in gymnosperms (e.g., approx. 600 *NBS-LRR* genes are present in *P. abies*; [79]). The origin of gene regulation by the miR482/2118 gene family may, therefore, be related to the expansion of the *NBS-LRR* gene family at the base of the seed plants. Altogether, the existing data suggest that the miR482/2118 gene family originated in the ancestor of seed plants possibly via the modification (sequence evolution) of an ancestral *NBS-LRR* gene.

The miR482/2118 family can be considered an ancient miRNA family because it is present across seed plants. In most ancient miRNA families, the nucleotide sequence of the functionally important mature miRNA region is highly conserved [80]. This is not the case for the miR482/2118 family. Instead, the mature region shows a substantial sequence divergence between distantly related plant lineages [61] and sometimes even between closely related species [73]. The difference may be related to the necessity of miR482/2118 family members to evolutionarily track their targets, which can evolve quickly [81,82,83]. Moreover, the coevolutionary tracking of the sequence divergence of the *NBS-LRRs* by the miR482/2118 members corresponds to the sequence variation across *NBS-LRR* genes present at the third codon positions of the targeted gene region encoding the P-loop [79]. This results in a lineage-specific repertoire of miR482/2118 variants that can target the wide diversity of *NBS-LRRs* expressed by these plant genomes.

The *NBS-LRR*-miR482/2118 regulatory network is also highly dynamic with respect to miRNA-targeting. Orthologous *NBS-LRR* genes between closely related taxa (for example, tomato and potato) are not necessarily targeted by orthologous miR482/2118 genes [73]. This implies that lineage-specific evolutionary changes in both (or either) the targets and/or miRNAs result in different predicted targeting relationships. Additionally, of the *NBS-LRR* genes predicted to be targeted by this miRNA family, a large proportion are targeted by multiple miRNAs [73]. This indicates that this transcriptional regulatory network displays a degree of robustness despite shifts in targeting. The transcriptional responses of orthologous miR482/2118 genes can also differ between closely related species, as is observed in the tomato clade. We observed that the expression of individual members of miR482/2118 genes differed across tomato species challenged by the same pathogen, *Phytophthora infestans* [66]. Given that the miRNA-mediated regulation of *NBS-LRRs* by miR482/2118 regulates plant immunity [67,77,78,84], differences in the expression of miR482/2118 members might also correspond to the observed differences in the progression of infection by *P. infestans* across different tomato species [66]. This demonstrates that, even on short evolutionary timescales (approx. 1 million years), the expression differences in miRNA genes can arise and may underlie observed differences in pathogen resistance between plant species.

The transcriptional regulation of *NBS-LRRs* in angiosperms is not exclusively controlled by the miR482/2118 gene family: Other lineage-specific miRNAs also target *NBS-LRRs* [79]. Whether other miRNA families contribute to the posttranscriptional control of resistance genes outside of the seed–plant lineage is not known. However, since the general machinery for the production of sRNAs is present throughout the green lineage [58], it is possible that this form of posttranscriptional regulation is employed to modulate biotic stress responses in all land plants and perhaps even in streptophyte algae.

## 4. Additional microRNAs Regulating Plant Immunity

The fine-tuning of plant immunity is not only achieved through the regulation of *NBS-LRRs* but also through the regulation of independent immune signalling pathways. Some of those immunity-associated pathways are also regulated by miRNAs [85,86]. These include the regulation of the auxin response factor (ARF)-encoding and other genes involved in auxin signalling by miRNA families miR160, miR167, and miR393 [85]. These three miRNA families are present across all vascular plants [87]. The two miRNA families miR159 and miR166 likely also arose in the common ancestor of vascular plants [87]. Both of these miRNAs are implicated as positive regulators of immunity to *Verticillium dahliae* in cotton [88]. Likewise, miR172, which targets the transcription factor-encoding gene *AP2*, was recently identified as a regulator of resistance to the oomycete *P. infestans* [89]. This miRNA family is also considered ancient and is conserved from ferns to angiosperms [87]. Hence, many miRNAs which have had a long evolutionary history in vascular plants and were originally associated with plant development may also have dual functions in plant immunity. These two types of miRNAs illustrate the significance of both young lineage-specific miRNAs and ancient, well-conserved miRNAs in regulating plant immunity through targeting both young and ancient gene regulatory networks.

## 5. The RNA Silencing Machinery as a Common Target of Pathogens

Pathogens and plants are in a relentless coevolutionary struggle with one another. The RNA silencing machinery, which is evolutionarily conserved across the green lineage [58], is a central regulatory mechanism in plants, and sRNAs play a crucial role in plant–pathogen interactions [86]. Effectors that target evolutionarily conserved components of immunity have evolved convergently in pathogens [21]. A common target is the transcription factor TEOSINTE BRANCHED, CYCLOIDEA AND PCF 14 (TCP14), which regulates immune signaling [20]. These *TCPs* appear to have originated in the common ancestor of land plants and streptophyte algae [33]—as TCPs are also found in the genome of *Chara braunii* [35,48]. Likewise, the RNA silencing machinery can also be hijacked by distinct pathogens. So-called RNA silencing suppressors are present and expressed by pathogenic viruses, bacteria, fungi, and oomycetes [10,11,12,90,91] (Figure 2). However, the pathogens do not convergently hijack an individual component of this machinery, but multiple steps of the process can serve as suitable targets (Figure 2). For example, while the effector HopT1-1 of *Pseudomonas syringae* interferes with AGO1 [10], the RNA silencing suppressor PSR1 of the oomycete *Phytophthora sojae* targets a novel but evolutionarily conserved component in the miRNA biogenesis machinery that affects the subnuclear localization of the dicing complex [92] (Figure 2). By manipulating the RNA silencing machinery, the pathogens are capable of broadly manipulating plant stress responses. However, since many pathogen suppressors of RNA silencing also lead to a reduction in miR482/2118 levels [61], they may inadvertently activate *NBS-LRRs*. Hence, miR482/2118-mediated RNA silencing may serve as a pathogen sensor in addition to its role in fine-tuning immunity [61,93].

While some RNA silencing suppressors are species-specific, others seem to be conserved across a wider range of related pathogens. A second RNA silencing suppressor described in the oomycete *P. sojae*, PSR2 [11], is such a case; it is present in several species of the genus *Phytophthora* [94]. Additionally, evolutionary analyses of PSR2 across several isolates of *P. infestans* reveal that this effector is conserved at the protein level [95]. Most of the detected amino acid variants were located in the so-called W- or Y-motifs [95], which are distributed throughout the PSR2 coding sequence and present in all PSR2-like sequences [95,96]. These motifs may allow for adaptive mutations while maintaining the overall protein structure and function [97]. In agreement with this, the first two WY-loops are essential for the virulence function of PSR2 in *Phytophthora capsica* [98].

The beneficial effect of PSR2 on virulence is conserved across different species of *Phytophthora* and hosts [94]. This observation, together with the observed protein sequence conservation, implies that the host target might also be a conserved and essential part of the RNA biogenesis machinery [95]. Indeed, Hou and colleagues [98] found in their recent study that PSR2 binds dsRNA-binding protein 4 (DRB4), which forms a complex with DCL4 to degrade dsRNA and generate phasiRNAs [99,100,101]. Hou and colleagues [98] provide evidence that the phasiRNAs generated during infection (of which the levels are reduced due to the action of PSR2) are not direct regulators of plant mRNAs but are instead sRNAs transported from the plant into *Phytophthora* (possibly via extracellular vesicles). Once inside the pathogen, these plant-derived sRNAs target a specific gene of the pathogen, the expression of which is required for infection success and sporulation [98]. This links the observations that PSR2 is a well-conserved virulence factor and targets a conserved host molecule to a reduction in the expression of host-derived sRNAs since these interfere with pathogen development.

To what degree this intricate counter-defense mechanism extends to other hosts interacting with *Phytophthora* sp. remains an open question. Thus far, this mechanism has been investigated only in angiosperm hosts of *Phytophthora* sp. However, species of *Phytophthora* can also infect nonflowering plants; for example, *Phytophthora palmivora* is capable of infecting the liverwort *M. polymorpha* [102]. The conservation of PSR2 across many species of *Phytophthora* [94,95] may suggest that a similar mechanism could be functional in a wide range of hosts. Yet, while the process of secondary sRNA biogenesis is present in bryophytes [103,104], the target of PSR2, DRB4, has not been found in the bryophyte *P. patens* or in the lycophyte *Selaginella moellendorffii*, although they both possess other DRBs [105]. Hence, the role of PSR2 during infection by *Phytophthora* sp. on non-angiosperm hosts remains to be elucidated. This is certainly a very exciting avenue of future research.

## 6. Conclusions

A hallmark of plant immunity is the swift and sweeping changes in the global transcriptional profile when a plant is challenged with a pathogen. Complex transcriptional networks have evolved in both interacting partners, and these allow for specific and appropriate responses. Across >900 MY of green evolution, many gene regulatory networks are conserved, at least to some degree, at the protein level—the degree of functional conservation remains to be explored. Here, we illustrated how these conserved regulatory networks are manipulated by pathogens to reduce plant defense responses and, thereby, increase the pathogen’s fitness. The exploitation of such gene regulatory networks allows pathogens to alter immune responses on a broad scale. Its effectiveness is apparent given that many pathogens independently target different components of the same transcriptional regulatory network, as was highlighted by the diversity of mechanisms pathogens use to manipulate the RNA silencing. The existence of the conserved effector PSR2, targeting a wide-spread component of the RNA-silencing machinery further underscores this observation. We can now use this information to learn more about the nature of the gene regulatory networks that govern plant immunity, as well as the evolutionary forces that shape those networks in response to pathogen attack.

## Figures and Tables

**Figure 1 genes-10-00310-f001:**
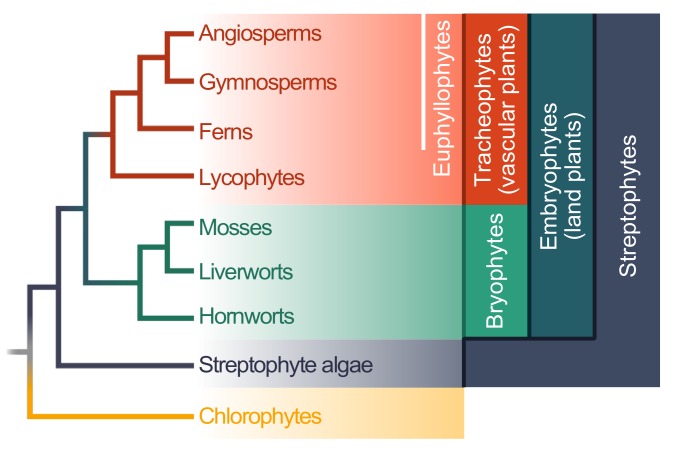
A simplified cladogram of the green lineage; The cladogram is based on a phylogeny recovered by Puttick et al. [23].

**Figure 2 genes-10-00310-f002:**
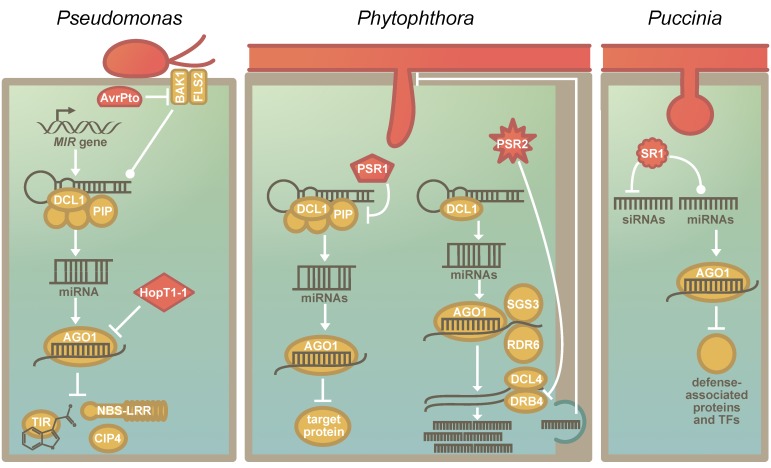
The suppression of the RNA silencing machinery of plants by pathogen-encoded RNA Silencing Suppressors: (**a**) The known effectors that affect host RNA silencing in *P. syringae*. AvrPto inhibits the association of BRASSINOSTEROID INSENSITIVE 1-associated receptor kinase 1 (BAK1) and FLAGELLIN-SENSITIVE 2 (FLS2), which results in the stabilization of pre-miRNAs and reduces miRNA abundance [10]. HopT1-1 interferes directly with the RNA silencing machinery by inhibiting the function of AGO1 [10]. Nucleotide-binding site-leucine-rich repeats (NBS-LRRs), TRANSPORT INHIBITOR RESPONSE 1 (TIR1) and COP1-interacting protein 4 (CIP4) are known targets of miRNAs that are affected by RNA silencing suppressors of *P. syringae.* (**b**) The known RNA silencing suppressors of *Phytophthora* sp. PSR1, a *P. sojae*-specific RNA silencing suppressor, interacts with PIP1. PIP1 is required for the correct assembly of the Dicing-bodies, which are the protein complexes that include DCL1. The destabilization of the Dicing-body results in reduced miRNA accumulation. PSR2 is conserved in several *Phytophthora* species. [94,95]. It binds and inhibits DRB4. DRB4 acts in the phasiRNA pathway, and PSR2 specifically reduces phasiRNAs that are secreted—likely via extracellular vesicles. These phasiRNAs have a target in *Phytophthora* and targeting impacts the virulence and sporulation of the pathogen [98]. (**c**) The function of the recently identified RNA silencing suppressor of *Puccinia graminis f.* sp. *tritici* (*Pgt*). *Pgt*SR1 (SR1) negatively regulates siRNA biosynthesis and affects miRNA biogenesis. The miRNAs that were affected had targets that are associated with plant defense signalling, including several defense-associated transcription factors [12].

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
