# Peer review of "The Elaboration of miRNA Regulation and Gene Regulatory Networks in Plant–Microbe Interactions"

_genes, 2019, doi:10.3390/genes10040310_

Reviewer 1 Report

The review ``The evolution and co-evolution of gene regulatory 2 networks in plant-microbe interactions`` by Vries et al is very timely and very well written. I do not see any major issues with the manuscript.

Author Response

AU: Thank you for this positive comment.

Reviewer 2 Report

The manuscript submitted by de Vries et al. describes the macroevolutionary aspects of plant immune signalling with a particular focus on micro-RNA mediated control of NBS-LRR genes and the pathogen effector proteins targeting these processes. My main issue is that this review is quite similar to the author’s previous review article focusing on the evolution of plant immune genes. Moreover, the manuscript contains some obvious errors that suggest a rushed submission. The missing text from Lines 133-135 and missing bars in Figure 1 (see below) are examples of this. The review, as it stands, does not significantly advance current hypotheses and research directions in the field, but I feel confident that the authors could address my concerns in a substantially revised version of the manuscript.

——————————————————————————————————————————

Major Concerns:

The title of the article suggests a broad focus on gene-regulatory networks during plant-microbe interactions yet there is little description of this in respect to well known defence signalling modules in angiosperms (for example WRKY, NAC, TGA families). Are these modules conserved, at least on the gene level, in bryophytes and other land plants? Instead, the authors focus on microRNA control of NBS-LRR genes (described in their previous review de Vries et al. 2018; Comm. Int. Biol.) and discuss pathogen effectors that target this process (a novel aspect worth describing here). Perhaps the title of the review should focus on this, or the authors can expand on their existing title within the main body of the review.

Lines 96-99: There is no good evidence to suggest that salicylic acid acts as a defence hormone in bryophytes. Partial complementation of npr1 or npr3/4 double mutants by strong constitutive overexpression of the Physcomitrella NPR homolog may only suggest that certain structural similarities allow for rescue in Angiosperms. Critically, we still lack any indication that SA or SA-like molecules are synthesized and perceived during immune responses in Bryophytes. Evidence may hint towards this, as described by a recent article focusing on the secretome of SA-treated moss (https://onlinelibrary.wiley.com/doi/abs/10.1002/psc.3138), but there is no foundational evidence for a conserved defence hormone pathway yet. 

Line 234 - TCPs are described as evolutionarily conserved targets of effectors in land plants but there is only evidence for this in angiosperms. Please clarify this in the manuscript.

——————————————————————————————————————————

Minor Concerns:

Throughout the manuscript the authors write ‘evolutionary’ when the word evolutionarily is the correct word. For example in section 3 on lines 231, 233, 241, etc. Please correct this where necessary throughout the manuscript.

Figure 1 — There appears to be some loss of quality in this figure during formatting. The descriptive tree/cladogram is not connected between groups of plant lineages and there are lines of varying thickness. If this is intentional then it would require an explanation in the legend, but this appears to be a formatting issue that can be easily fixed. 

Line 83 - A. thaliana should be written as Arabidopsis thaliana at first use. 

Lines 105-107:  Are you referring to a single RLK here (if so, which one?), or the family in general. If referring to the family then clearly state this.

Line 128 - Please provide sufficient context for the placement of Amborella in this position for readers not familiar with land plant evolution.

Author Response

The manuscript submitted by de Vries et al. describes the macroevolutionary aspects of plant immune signalling with a particular focus on micro-RNA mediated control of NBS-LRR genes and the pathogen effector proteins targeting these processes. My main issue is that this review is quite similar to the author’s previous review article focusing on the evolution of plant immune genes. Moreover, the manuscript contains some obvious errors that suggest a rushed submission. The missing text from Lines 133-135 and missing bars in Figure 1 (see below) are examples of this.

AU: Thank you for noticing these errors. We apologize for this. Both of these errors emerged after the further processing of the pdf to be sent out to the reviewers and were not visible in the pdf version we approved.  The text on line 133/134 is a header for a new section, which starts in line 135. In this line the first word is supposed to be “Biotic” not “iotic”. We hope this issue will not arise during the next pdf conversion. 

The review, as it stands, does not significantly advance current hypotheses and research directions in the field, but I feel confident that the authors could address my concerns in a substantially revised version of the manuscript.

——————————————————————————————————————————

Major Concerns:

The title of the article suggests a broad focus on gene-regulatory networks during plant-microbe interactions yet there is little description of this in respect to well known defence signalling modules in angiosperms (for example WRKY, NAC, TGA families). Are these modules conserved, at least on the gene level, in bryophytes and other land plants? Instead, the authors focus on microRNA control of NBS-LRR genes (described in their previous review de Vries et al. 2018; Comm. Int. Biol.) and discuss pathogen effectors that target this process (a novel aspect worth describing here).

AU: We thank the reviewer for the interest in our current and previous work. We agree that the example of miR482/2118-based regulation of NBS-LRRs was also part of our previous review. However, in the previous work we focused more on the evolution of the NBS-LRRgenes, while in this review we concentrated on the evolution of miRNA targeting. In this manuscript, we also review how targeting can evolve and how this shapes the gene regulatory network. This is a perspective we have not considered previously. Moreover, since this area of research is very active, additional papers have come out in the last year that were not available at the time of our publication in Comm. Int. Biol. For example, it was recently reported that a member of the miR482/2118 family in tomato is expressed in reproductive tissue. This changes our interpretation on the timing and degree of the specialization of the miR482/2118 family on NBS-LRRgenes. We believe it was important to integrate these new insights in the current manuscript. Furthermore, we did not discuss the evolutionary forces that shape this regulatory network in our previous data. However, to increase the novelty of this manuscript we have included a new subsection on other evolutionarily conserved miRNA families that do not regulate NBS-LRRs but also play important roles in plant immunity.

Perhaps the title of the review should focus on this, or the authors can expand on their existing title within the main body of the review.

AU: We agree with suggestion of the reviewer and changed our title. The new title is: ‘The elaboration of miRNA regulation and gene regulatory networks in plant-microbe interactions’

Lines 96-99: There is no good evidence to suggest that salicylic acid acts as a defence hormone in bryophytes. Partial complementation of npr1 or npr3/4 double mutants by strong constitutive overexpression of the Physcomitrella NPR homolog may only suggest that certain structural similarities allow for rescue in Angiosperms. Critically, we still lack any indication that SA or SA-like molecules are synthesized and perceived during immune responses in Bryophytes. Evidence may hint towards this, as described by a recent article focusing on the secretome of SA-treated moss (https://onlinelibrary.wiley.com/doi/abs/10.1002/psc.3138), but there is no foundational evidence for a conserved defence hormone pathway yet.

AU: We thank the reviewer for pointing this out. We agree that overexpression can have ectopic effects and that these experiments alone do not present bullet-proof evidence about the actual function of NPR1 in bryophytes. We have now added a sentence noting that additional experiments are required to elucidate the conservation of the function of NPR1 with respect to immunity in bryophytes. We, however, would also like to point out that SA was reported to accumulate after pathogen attack in Physcomitrella patensand also induces gene expression of genes related to defense against necrotrophic pathogens (see for example Ponce de León et al. 2012 and Oliver et al. 2009), which is a first hint that SA functions at least in immunity signaling of mosses.

Line 234 - TCPs are described as evolutionarily conserved targets of effectors in land plants but there is only evidence for this in angiosperms. Please clarify this in the manuscript.

AU: We apologize for this confusion. We wanted to say that TCP14 is a common target of effectors from different pathogens and that TCPs are evolutionarily conserved from streptophyte algae to land plants. It might therefore be that pathogens target effectors that are well embedded in signaling networks, due to their long evolutionary history. We have now rewritten this sentence: ‘Effectors that target evolutionarily conserved components of immunity have evolved convergently in pathogens [21]. A common target is the transcription factor TEOSINTE BRANCHED, CYCLOIDEA AND PCF 14 (TCP14), which regulates immune signaling [20]. These TCPs appear to have originated in the common ancestor of land plants and streptophyte algae [33].’

——————————————————————————————————————————

Minor Concerns:

Throughout the manuscript the authors write ‘evolutionary’ when the word evolutionarily is the correct word. For example in section 3 on lines 231, 233, 241, etc. Please correct this where necessary throughout the manuscript.

AU: We have corrected this.

Figure 1 — There appears to be some loss of quality in this figure during formatting. The descriptive tree/cladogram is not connected between groups of plant lineages and there are lines of varying thickness. If this is intentional then it would require an explanation in the legend, but this appears to be a formatting issue that can be easily fixed.

AU: Thank you for bringing this to our attention. This is indeed a conversion error that was not visible in the final PDF proof we received for approval. We were able now to access the final PDF file the reviewers could download. We hope that in the next pdf conversion this error will not occur again. If so, we will contact the editorial office to correct this mistake before review.

Line 83 - A. thaliana should be written as Arabidopsis thaliana at first use.

AU: Thank you for noticing. We have amended this.

Lines 105-107:  Are you referring to a single RLK here (if so, which one?), or the family in general. If referring to the family then clearly state this.

AU: In the study from Delaux et al. 2015 only one RLK per species was identified in the transcriptomes of several streptophyte algae. Later on Nishiyama et al. 2018 found an independent expansion of RLKs in Chara braunii(see p.5 l.118 – p.6 l.125). Hence the current assumption is that the last common ancestor of land plants and streptophyte algae presumably had one ancestral RLK which underwent independent expansions in the two lineages. To clarify this we have added the following information: ‘… This study [52] also found one receptor-like kinase (RLK) in the transcriptomic data of several streptophyte algae’

Line 128 - Please provide sufficient context for the placement of Amborella in this position for readers not familiar with land plant evolution.

AU: We have added the following information to this sentence: “… and Amborella(the sister-lineage to all angiosperms)and angiosperms”

Reviewer 3 Report

I liked this review very much, I found it a pleasure to read and very informative. I have a few minor suggestions for improvement.

In order to facilitate that the reader can follow all miRNA-related arguments, it would be very helpful to present an additional figure that reviews the different RNAs and RNA-related defense mechanisms and where the known effectors impact on these mechanisms.

Line 111, 203, and 208-209: lineage-specific

Lines 134-135 here is something missing

Line 170 explain what is meant by miRNA*

Line 210 NBS-LRR genes

Line 222 is not exclusively

Line 232 plants, and

Line 233 and 241: evolutionarily conserved

Line 233 of immunity

Line 234 replace TF by transcription factor

Line 273 the role of PSR2

Author Response

I liked this review very much, I found it a pleasure to read and very informative. I have a few minor suggestions for improvement.

AU: Thank you for this positive comment.

In order to facilitate that the reader can follow all miRNA-related arguments, it would be very helpful to present an additional figure that reviews the different RNAs and RNA-related defense mechanisms and where the known effectors impact on these mechanisms.

AU: We have added an additional figure on the miRNA signaling and pathogen interference to the manuscript.

Line 111, 203, and 208-209: lineage-specific

AU: We have corrected this.

Lines 134-135 here is something missing

AU: Thank you for bringing this to our attention. We apologize for this. The final PDF proof we got to check seems different from the final PDF file the reviewers received. Line 133/134 was a different header section and in line 135 the first word is supposed to be “Biotic” not “iotic”. We will contact the editorial office to correct this mistake and hope this issue will not arise again. 

Line 170 explain what is meant by miRNA*

AU: We have added the following explanation: ‘… specifically in regions outside of the miRNA/miRNA* region [75]. The miRNA/miRNA* is the dsRNA region excised from the pre-miRNA that includes the functional miRNA itself and its complementary sequence denoted as miRNA*.’’

Line 210 NBS-LRR genes

AU: We have corrected this.

Line 222 is not exclusively

AU: We have amended this.

Line 232 plants, and

AU: We have added the comma.

Line 233 and 241: evolutionarily conserved

AU: We have corrected this

Line 233 of immunity

AU: We have corrected this mistake.

Line 234 replace TF by transcription factor

AU: We replaced TF with transcription factor.

Line 273 the role of PSR2

AU: We have corrected this.

Round  2

Reviewer 2 Report

The authors have addressed all of my concerns and I can now endorse their manuscript for publication.